# Novel Experimental Mouse Model to Study the Pathogenesis and Therapy of *Actinobacillus pleuropneumoniae* Infection

**DOI:** 10.3390/pathogens13050412

**Published:** 2024-05-15

**Authors:** Duc-Thang Bui, Yi-San Lee, Tien-Fen Kuo, Zeng-Weng Chen, Wen-Chin Yang

**Affiliations:** 1Agricultural Biotechnology Research Center, Academia Sinica, Taipei City 115, Taiwan; ducthangbui.hup@gmail.com (D.-T.B.); jj61226@yahoo.com.tw (Y.-S.L.); tienfen@gate.sinica.edu.tw (T.-F.K.); 2Institute of Biotechnology, National Taiwan University, Taipei City 106, Taiwan; 3Animal Technology Research Center, Agricultural Technology Research Institute, Miaoli County 350, Taiwan; zwc@mail.atri.org.tw; 4Department of Life Sciences, National Taiwan Ocean University, Keelung City 202, Taiwan; 5Graduate Institute of Integrated Medicine, China Medical University, Taichung City 404, Taiwan; 6Department of Life Sciences, National Chung-Hsing University, Taichung City 404, Taiwan

**Keywords:** *Actinobacillus pleuropneumoniae*, clinical score, inflammation, lung, mouse model, pleuropneumonia, pulmonary bacterial infection

## Abstract

*Actinobacillus pleuropneumoniae* (APP) is a major cause of lung infections in pigs. An experimental mouse has the edge over pigs pertaining to the ease of experimental operation, disease study and therapy, abundance of genetic resources, and cost. However, it is a challenge to introduce APP into a mouse lung due to the small respiratory tract of mice and bacterial host tropism. In this study, an effective airborne transmission of APP serovar 1 (APP1) was developed in mice for lung infection. Consequently, APP1 infected BALB/c mice and caused 60% death within three days of infection at the indicated condition. APP1 seemed to enter the lung and, in turn, spread to other organs of the mice over the first 5 days after infection. Accordingly, APP1 damaged the lung as evidenced by its morphological and histological examinations. Furthermore, ampicillin fully protected mice against APP1 as shown by their survival, clinical symptoms, body weight loss, APP1 count, and lung damages. Finally, the virulence of two extra APP strains, APP2 and APP5, in the model was compared based on the survival rate of mice. Collectively, this study successfully established a fast and reliable mouse model of APP which can benefit APP research and therapy. Such a model is a potentially useful model for airway bacterial infections.

## 1. Introduction

Swine respiratory diseases have considerable economic impact on intensive pig farming worldwide [1]. The major causative bacteria of respiratory diseases include the genus of *Actinobacillus*, *Mycoplasma*, *Haemophilus*, and *Streptococcus* [2,3]. These infectious pathogens can cause pneumonia, pleuritis, and/or pleuropneumonia in pigs [4]. Among them, *A. pleuropneumoniae* (APP) is a Gram-negative bacterium that does not form spores [5] and is facultatively anaerobic. APP is one of the most severe bacterial pathogens in the lung of swine. It causes swine pleuropneumonia in pigs of all ages and incurs a significant economic loss in the pig industry [6,7]. Now, 19 serovars of APP, including APP1 to APP19, have been documented. APP1 and APP5 have been reported to be highly virulent strains, whilst APP2 was a lowly virulent strain. Antibiotics and vaccines are commonly used to treat APP and other bacterial infections in the lung of animals [5]. Intravenous injection of antibiotics is a more convenient route than oral consumption. However, the oral administration of antibiotics needs to consider some extra factors, including the absorbability of antibiotics from the gut to the circulation, the binding of antibiotics and plasma proteins, and the ability to reach the lung. Moreover, antibiotic resistance poses a global threat pertaining to lung infections, so novel methods to screen effective anti-bacterial drugs with novel mechanisms are needed [4]. APP that resists one and multiple antibiotics have been found in pig farms though APP is not a human pathogen [5,8]. Even in cases of no antibiotic resistance and residue, vaccines for APP are sometimes ineffective because of a lack of appropriate antigens, low cross-reactivity, and immune escape.

Animal models of pulmonary bacterial infections are extremely important for the etiology of the lung disease, causative mechanisms, drug discovery and development, and therapeutic efficacy assessment of anti-bacterial drugs [9,10]. Mouse models are the most common animal models of lung infections because of their short life, small body size, high physiological similarity with other mammals, sufficiency of genetic information, and ease of conducting various experiments [9,11]. Despite these numerous merits, the narrowness of the respiratory tract and pathogen-to-host specificity in mice make it challenging to establish a murine model of bacterial infections in the lung. To overcome the hurdles, intraperitoneal injection and nasal instillation have been adopted to infect mice with bacteria [11,12,13,14]. At first glance, the intraperitoneal injection of bacteria seems to be an easier method. However, a big drawback of this model is that injection causes a systemic infection in mice and, indeed, is etiologically not limited to the respiratory system [9,15]. Nasal instillation infects mice with bacteria in a more natural way. However, a technical caveat of this method is that the nasal cavities of the mice are so narrow that it is difficult to administer liquid drops of bacteria so that they consistently enter both cavities of the lung of mice, and they frequently leak out of the nasal cavities but not the other around. Furthermore, murine nasal cavities and windpipes are too thin to perform nasal injections, and furthermore, mouse anesthesia is a requirement [14].

The purpose of this study was to establish a fast and reliable mouse model of APP for disease study and therapy. Here, an aerosol-mediated lung infection method was developed to infect BALB/c mice with APP. Next, the pathology of APP1-infected mice was characterized. The therapeutic efficacy of ampicillin in APP1-infected mice was also assessed. Finally, the mortality of additional APP serovars, APP2 and APP5, with low and high virulence, respectively, was compared in mice.

## 2. Materials and Methods

### 2.1. Chemicals, Reagents, Bacterial Strains, and Animals

Brain heart infusion (BHI, BD Difco, Franklin Lakes, NJ, USA) and β-nicotinamide adenine dinucleotide (NAD, Sigma–Aldrich, Burlington, MA, USA) were purchased. Different serotypes of APP, including APP1 (ATCC 27088), APP2 (ATCC 27089) and APP5 (ATCC 33377), were used in this study. Different APP strains were grown in BHI agar plates containing NAD (10 μg/mL) at 37 °C overnight. One colony of each strain was grown in 5 mL BHI medium containing NAD medium for 16 h, and the optical density (OD) of the colonies at 600 nm equaled 1. The bacteria were amplified in BHI medium at a 1 to 50 ratio until the OD_600_ reached 0.6. Then, the bacteria were counted and diluted to 5 × 10^10^ CFU per ml for mouse infection. Thirty-five 6- to 8-week-old, male BALB/c mice, weighing 20–22 g, were purchased from National Laboratory Animal Center (Taipei City, Taiwan). All animals were fed with standard diets kept at 22 ± 1 °C with 12 h dark–12 h light cycles in the institutional animal facility and handled in compliance with the Academia Sinica Institutional Animal Care and Utilization Committee (2111-1739) guidelines.

### 2.2. Setup and Study of Pulmonary Bacterial Infection in Mice

An aerosol system controller, a nebulizer head, a nebulizer control unit, and an acrylic chamber were assembled to generate aerosols for lung infection (Figure 1). Bacterial aerosol droplets with a size of 4 to 6 μm were generated by vibration and conducted into an air-tight chamber for aerosol exposure in order to infect the lung of mice. Briefly, 1 mL of PBS in the absence or presence of bacteria was vibrated into aerosol droplets for 5 min using a Buxco nebulizer (Data Sciences International, St. Paul, MN, USA), and the aerosol droplets were exposed to mice for an additional 20 min. The working conditions were set as follows. The nebulizer flow rate was set to 0.4 mL/minute, the bias flow was set to 2.4 L per minute, and the duty cycle was set to 63%. PBS (1 mL) was loaded to the Buxco nebulizer head. Mice were constantly restrained in the chamber filled with the aerosols for 20 min. After inhalation, the fur of the mice was disinfected with 70% isopropanol wipes. Using aerosol-mediated APP1 infection, BALB/c males were randomly divided into 3 groups, 5 mice in a group. On day 0, one group of mice (NC) was exposed to PBS aerosols, the second group (APP1) was exposed to APP1 aerosols (5 × 10^10^ CFU), and the third group (APP1 + AMP) was exposed to the same amount of APP1 aerosols, followed by an intraperitoneal injection of ampicillin (20 mg/kg) at 2 h post-infection. Each group of mice was monitored daily for gross and microscopic examinations. Alternatively, mice were infected with APP2 and APP5, and the same parameters were monitored. All animals were maintained at 21–23 °C with 12 h light–12 h dark cycles in the institutional animal facility and handled according to the Academia Sinica Institutional Animal Care and Utilization Committee (AS-IACUC 21-11-1739).

### 2.3. Calculation of Mortality, Clinical Score, and Body Weight

After bacterial lung infection, mice were monitored daily for mortality, clinical score, and body weight loss for a period of over 5 days. A clinical score was obtained from the summation of the activity, fur, appetite, mental status, and eye secretions of the mice as described in [13]. The clinical score assessment criteria were modified as previously described in [13]. Body weight loss (%) was obtained from the formula, and body weight loss (%) = 100% × (1 − (BWday x/BWday 0)).

### 2.4. Bacterial Counting and Histochemical Staining of Mouse Organs

After sacrifice, the lung, spleen, liver, kidney, and heart of 3 mice per group, at 1-, 2-, 3-, 4-, and 5-day post-infection, were removed, photographed and weighed as described in [11]. One part of these organs was weighed, homogenized, and counted for bacterial load [11]. Briefly, organ slices were added into the tube with sterile PBS in a ratio of 1:10 (g/mL), ground with the taco™Prep Bead Beater (GeneReach Biotechnology Co., Taichung City, Taiwan), serially diluted in PBS, and spread on BHI agar plates supplemented with 10 μg/mL NAD for growth at 37 °C. After 24 h, the bacteria on the plates were counted.

The other part was fixed with 10% formalin for 48 h, followed by histochemical staining. Briefly, the fixed lung was embedded with paraffin, sectioned with a microtome (Leica TP 1020), and stained with hematoxylin and eosin. The slides were examined under a microscope, photographed, and analyzed.

### 2.5. Statistics

Results from three or more independent experiments are presented as mean ± standard deviation. One-way ANOVA test was used for the analysis of statistical differences among the means of two or more groups in clinical score, body weight, and bacterial loads. The log rank test was applied to test the null hypothesis that there is no difference between the populations in the probability of death at any time point. *p* * < 0.05; *p* ** < 0.01; and *p* *** < 0.001 are considered statistically significant.

## 3. Results

### 3.1. Invention of a Fast and Reliable Mouse Model Using Aerosol-Mediated Bacterial Infection in Lung

Lung infections are the top causes of death in humans and animals. Mouse models of pulmonary bacterial infections are useful for research on disease etiology and therapy but have the drawbacks of anatomical narrowness of airways and host tropism. To facilitate easier and more convenient experimental bacterial infection of mice, in this study, an aerosol-mediated lung infection model in BALB/c mice was developed as described in Figure 1. The aerosol system controller (Figure 1a) was connected to a nebulizer head via a nebulizer control unit (Figure 1b) as a whole apparatus (Figure 1c). The nebulizer head comprised a cup, a piezoelectric element, and an aperture plate with holes, onto which 1 mL of bacterial suspension was added. After applying electricity, the aerosol system controller caused the plate to vibrate, creating a micro-pumping action that generated fine particle aerosols measuring 4 to 6 μm in size. For the aerosol transmission of pathogens to BALB/c mice, APP1 was used as an exemplary bacterial pathogen, followed by an aerosol exposure of APP1 to the mice for 25 min in an air-tight chamber. This aerosol-mediated lung infection method was proved to be a convenient and highly reproducible means of studying pulmonary bacterial infections in mice.

### 3.2. Pathology of APP1-Infected Mice

The aerosol-mediated lung infection in mice was applied as delineated in Figure 1. The study involved three groups of mice, including a negative control group exposed to PBS aerosol droplets (NC), an infected group exposed to APP1-containing PBS aerosol droplets (APP1), and an infected group treated with ampicillin (positive control (PC)). Day 0 was an infection day. Two hours after infection, mice in the PC group were injected with Amp 20 mg/kg BW intraperitoneally. A variety of parameters in three groups were measured within 5 days post-infection.

As expected, the control mice survived for 5 days (NC, Figure 2a). In sharp contrast, mice infected with APP1 started to die from day 1 post-infection. As a result, the APP1 group had a 5-day survival rate of 40% (APP1, Figure 2a). Accordingly, mice with APP1 infection had a higher clinical score than that of the control mice (APP1 vs. NC, Figure 2b) as evidenced by motion, clothing hair, appetite, mental status, and eye secretions. As expected, no clinical signs were observed in the control group (NC, Figure 2a). Furthermore, mice with APP1 infection showed body weight loss while the control mice did not (APP1 vs. NC, Figure 2c).

The bacterial loads of different organs (the lung, spleen, liver, heart, and kidney) of the mice were measured at one-day intervals within 5 days post-infection. As expected, no bacteria were detected in the organs of mice in the NC group (NC, Figure 3). The results revealed that the bacterial number of the five organs of the APP1 group was significantly higher than that of the control group (APP1 vs. NC, Figure 3). Notably, the bacterial count reached a plateau in the lungs of the mice with APP1 infection at 2 days post-infection and then declined over time (APP1, lung, Figure 3a). In parallel, the number of bacteria peaked at 3 days post-infection in the mouse spleen, liver, heart, and kidney and, then, decreased over time (Figure 3b–e). The data suggested that APP1 first invaded the lungs of the mice and then spread to their other organs via blood.

Gross and microscopic examinations of the lung of mice were performed. Macroscopically speaking, the morphological data showed that the mice with APP1 infection had more severe pulmonary hemorrhage and swelling than non-infected control mice. This severity in the mouse lungs reached a plateau within 2 days post-infection and gradually declined over time. As expected, no lesions were observed in the lungs of the mice in the control mice.

Microscopically, APP1-infected mice had more leukocyte infiltration in their lungs and lung lesions. Accordingly, their leukocyte accumulation and lung lesions reached a plateau within 2 days post-infection and then recovered over time. However, no leukocyte accumulation and lesions in the lungs of non-infected mice were seen.

The details of the data on ampicillin treatment in mice are described in Section 3.2 of the Results Section.

### 3.3. Therapeutic Assessment of Ampicillin in APP1-Infected Mice

The therapeutic effect of ampicillin on the survival rate, clinical score, body weight loss, number of bacteria in organs, and lung damages was also assessed. Consequently, one shot of 20 mg/kg ampicillin fully rescued the mice from death compared to APP1-infected mice (APP1 vs. PC, Figure 2a). Furthermore, ampicillin improved the clinical score and body weight loss of infected mice (APP1 vs. PC, Figure 2c) and reduced the APP1 number in the lungs as well as other organs of the mice (APP1 vs. PC, Figure 3). Accordingly, APP1-infected mice, which were treated with ampicillin, had less severe hemorrhage and swelling than non-medicated APP1-infected mice. Furthermore, APP1-infected mice, which were treated with ampicillin, significantly reduced the leukocyte infiltration and lung lesions in their lungs compared with non-medicated APP1-infected mice. As a result, ampicillin treatment had better recovery in APP1-infected mice based on lung pathogenesis in mice. Taken together, this model was feasible for research and development of therapy against APP1.

### 3.4. Comparison of Pathogenic Virulence in APP-Infected Mice

To assess the applicability of this model to other APP strains, the same mouse model was utilized to compare the virulence of APP2, a low-virulent strain, and APP5, a high-virulent strain. The survival rate of the mice infected with APP2 and APP5 at 5 × 10^10^ CFU was 80% and 20% (APP2 vs. APP5, Figure 4). Notably, the survival rate was lower in the mice with APP5 infection than in the mice with APP2 infection. Surprisingly, APP5 (Figure 4) had a lower survival rate in BALB/c mice than APP1 (Figure 2). The mice were more susceptible to APP5 and APP1 than APP2 (Figure 4). Taken together, an aerosol-mediated lung infection method for APP study in mice was successfully established.

## 4. Discussion

APP, a causative pathogen of porcine pleuropneumonia, jeopardizes pig health and results in a significant economic loss in the swine industry worldwide [5]. Airborne transmission is a critical mechanism for the spread of swine APP. However, etiological and mechanistic studies of APP in swine are costly. Mice are widely acceptable animal models to study the cause, pathogenesis, and therapeutic assessment of swine APP, owing to their advantages of body size, reproductive cycle, cost, handling convenience, and high similarity of physiology to other mammals [9,10,11]. Although several mouse models of APP have been reported [11,12], these models have detrimental flaws for the study of this disease probably because of technical hurdles of infecting mice via narrow respiratory tracts in mice. In this study, a superiority of aerosol-mediated APP infection in BALB/c mice was demonstrated. First, using this method, infection of mice via a natural respiratory route was easy and stable. In addition, inhalation of aerosols is physiological and more desirable than (intra)nasal instillation in mice. Second, such infection was more specific for lung infection than the intraperitoneal injection method. Intraperitoneal injection routes are primarily used to induce systemic pneumococcal infections that lead to bacteremia and possible secondary pneumonia. Mice intraperitoneally infected with bacteria appear to show bacterial spread from the peritoneal cavity to the circulation, which does not accurately reflect the mechanism of APP infection in the lungs. Third, aerosolized APP has even access to the lung compared to nasal instillation, and consistent experimental outcomes can be obtained. Due to the anatomical constraint of the nasal cavity, mice need to be anesthetized before the nasal administration to respiration system. Previous studies have shown that nasal instillation significantly influenced the resulting morbidity and mortality following infection [16]. Moreover, intranasal instillation in mice was technically and physically difficult in such narrow airways of mice and needed to be conducted in anesthetized mice to avoid a nasal reflex. Moreover, the anatomical location and connectivity of the nose to other organs might raise concerns about the deposition extent of bacteria in the lung since rather large amounts might be retained in the nose or reach the digestive organs [14]. Finally, APP in mice underwent a similar transmission route as pigs though mice are not the natural hosts of this bacterium [5]. This study proved the feasibility of using aerosolized pathogens to infect mice to establish a universal mouse model of lung bacterial infections using the aerosol exposure system.

Here, the Buxco aerosol system was used to spray 4 to 6 μm aerosol particles containing bacteria in an air-tight chamber where mice were exposed to the airborne pathogen-carrying particles for a given time (Figure 1). APP1, known as a prevalent and virulent strain in pigs worldwide, was chosen to infect BALB/c mice. The data revealed that APP1 at 5 × 10^10^ CFU, exposed for 25 min, caused a fatality of 60% in mice with severe symptoms such as crouching, piloerection, comatose, apastia, and reduced body weight (Figure 2b,c). The data suggested that mice were exposed to APP1-containing aerosol (particles of 4–6 μm) successfully. Exposure time is vital for aerosol infection [17]. Twenty-five min was appropriate for mice to inhale a sufficient dose of potentially fatal bacteria. The clinical symptoms of mice in this study (Figure 2b) are comparable to those observed in mice that were infected intranasally or intraperitoneally in previous experiments [11,12,13,18]. Also, the infection sites and timing in mice and pigs are similar (Figure 2, Figure 3 and Figure 4). After APP1 infection, APP was initially found in the lungs of mice within 2 days post-infection. It took 3 days to reach the spleen, liver, and other organs, i.e., liver and kidney of mice (Figure 3). The number of APP in the liver and spleen was higher than other organs because bacterial clearance mainly occurred there [19]. The data are consistent with the infection of other lung bacteria, including *M. tuberculosis* [20,21], *Burkholderia pseudomallei* [22], *M. abscessus* [23], and *Pseudomonas aeruginosa* [24]. From a histological perspective, mice with an aerosol-mediated APP infection developed similar lung pathogenesis as those infected with APP via intranasal or intraperitoneal administrations in terms of alveolar rupture, pulmonary congestion, and the infiltration of inflammatory cells into the lungs [11,12,13,18,25]. Furthermore, this model was suitable for therapeutic evaluation as demonstrated by ampicillin at the dose of 20 mg/kg (Figure 2, Figure 3 and Figure 4).

Both inhalation of aerosols and (intra)nasal instillation are in vivo methods that introduce APP to the mouse lung. Both have strengths and weaknesses. Aerosol exposure can be considered preferable to (intra)nasal instillation exposure for the following reasons. One key advantage of inhalation of aerosols is that it is the physiological route of exposure to airborne pathogens. Inhalation of aerosols could therefore cause fewer tissue lesions and inflammation than an (intra)nasal instillation procedure. Despite numerous advantages of the aerosol-mediated APP infection in mice, some limitations are seen in this model. Notably, APP causes lung bleeding, pneumonia, and pleuropneumonia in pigs [5]. However, less severe symptoms in the infected mice were observed. Nose bleeding and pleuropneumonia were not seen in APP-infected mice probably due to a lack of severe lung damage (Figure 2). Although lipopolysaccharide and other virulent factors might have similar receptors in mice and pigs, it was reported that the receptor of Apx toxin, β2-integrin, in pigs was functionally different from its murine homologue. This may explain the difference in the severity of lung pathogenesis between mice and pigs as well as that in the effective dosage of APP1 for both animals. Nevertheless, in this mouse model, a distinction could be made in virulence among APP strains based on mortality (Figure 2). As such, this kind of model can be used as an in vivo test tube to develop prophylactic and therapeutic remedies for APP. Furthermore, this model is practically useful for the investigation of bacterial infections in the airways of animals and humans. This study is the first study using aerosol droplets to infect mice with airborne bacteria. The findings serve as a cornerstone for the study of pathophysiology of respiratory bacteria and their therapeutics.

## 5. Conclusions

A BALB/c mouse model of APP infection was successfully established by aerosol inhalation. The aerosol-mediated infection in mice more closely resembles a natural infection in hosts than the previously established methods in terms of infection route and pathological changes in the lungs. As a result, this model can assist the investigation of the etiology, pathology, and therapeutic assessment of APP in mice. Notably, the same model is applicable to other airway bacterial infections.

## Figures and Tables

**Figure 1 pathogens-13-00412-f001:**
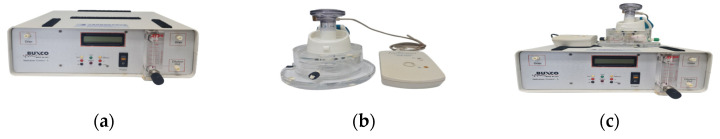
An apparatus for aerosol-mediated lung infection in mice. Aerosol system controller (**a**) and air-tight chamber with nebulizer head connected with the nebulizer control unit (**b**) are assembled into an aerosol generation and infection device (**c**).

**Figure 2 pathogens-13-00412-f002:**
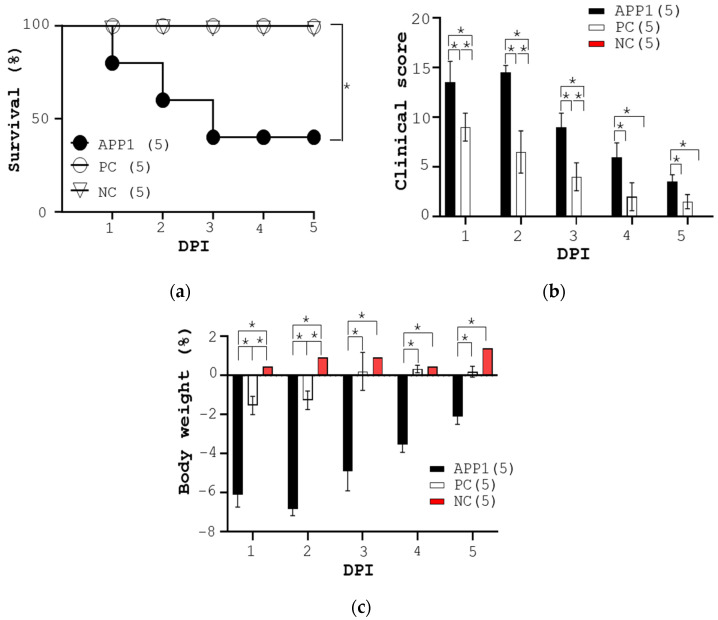
The survival rate, clinical scores, and body weight loss of the mice were measured. (**a**–**c**) Three groups of mice, 5 animals per group, were randomly assigned. One group of mice (NC) was exposed to PBS aerosols, the second group (APP1) was exposed to APP1 aerosols (5 × 10^10^ CFU), and the third group (PC) was exposed to the same amount of APP1 aerosols, followed by an oral gavage of ampicillin (Amp, 20 mg/kg) at 2 h post-infection. Each group of mice was monitored daily for survival rate (**a**), clinical score (**b**) and percentage of body weight (**c**). *p* * < 0.05.

**Figure 3 pathogens-13-00412-f003:**
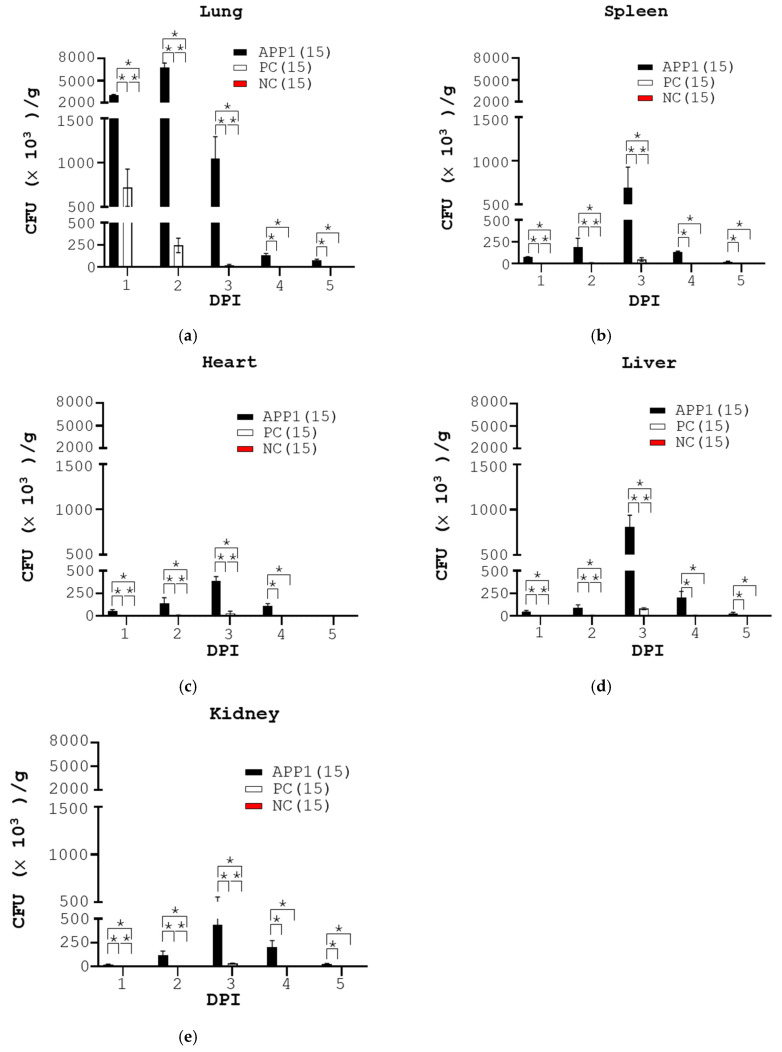
Bacterial count of different organs of the mice over 5 days following APP1 infection. The mice that received the same treatments as those in Figure 2 were sacrificed, three mice per group, at one to five days post-infection (DPI). The colony-forming unit (CFU) per gram of the lung (**a**), spleen (**b**), heart (**c**), liver (**d**), and kidney (**e**) from the mice was counted. *p* * < 0.05.

**Figure 4 pathogens-13-00412-f004:**
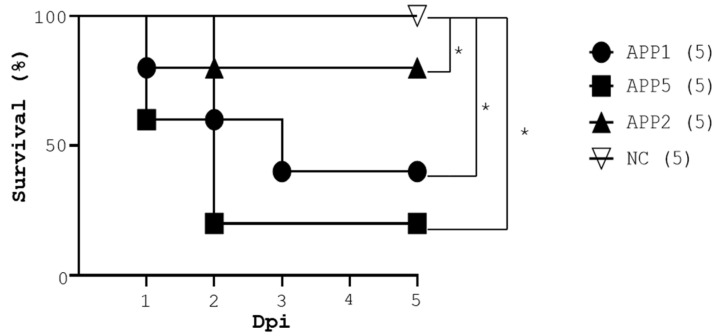
Comparison of the survival rate of BALB/c mice infected with APP5 and APP2. Three groups of mice, 5 animals per group, were randomly assigned. One group of mice (NC) was exposed to PBS aerosols, the second group (APP2) was exposed to APP2 aerosols (5 × 10^10^ CFU), and the third group (APP5) was exposed to the same amount of APP5 aerosols. Each group of mice was monitored daily for survival rate. The mouse number is shown in the parenthesis. *p* * < 0.05.

## Data Availability

The raw data supporting the conclusions of this manuscript are included in this manuscript.

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
