# Peer review of "Novel Experimental Mouse Model to Study the Pathogenesis and Therapy of Actinobacillus pleuropneumoniae Infection"

_pathogens, 2024, doi:10.3390/pathogens13050412_

Round 1

Reviewer 1 Report

Comments and Suggestions for Authors

The manuscript aims to develop a new model of respiratory infection in mice infected with Actinobacillus pleuropneumonia instead of pigs. Experimenting with pigs is costly, so this research provides valuable information. Overall, the manuscript is well-structured and written. Please refer to the following items to create a more complete manuscript.

Point-1. L107-114. Describes the number of mice in a group. How was the APP1 aerosol calculated with 5 × 10^10 CFU?

Why did you expose the APP in the aerosol to ampicillin (APP1+AMP) before infecting it? Did you confirm the toxicity or pathogenic genes of the APP before inoculation?

Point-2. 2.5. The statistics need to be explained in more detail.

Point-3, the authors used aerosol-mediated lung infection at L160. Why didn't they use direct inoculation into the lungs? Direct inoculation can have precise bacterial numbers.

Point-4. 3.3. The authors evaluated the therapeutic efficacy of ampicillin in mice infected with APP1. Did they confirm in vitro susceptibility tests such as MIC or MBC of ampicillin? The authors suggested that this model is suitable for research and development of treatments for APP1.

Point-5. L235, mice were more susceptible to APP5 and APP1 than to APP2. To confirm the results, it would be advisable for the authors to verify the expression of virulence genes such as ApxI, ApxII, and ApxIII.

Point-6. If possible in the discussion section, please compare the method of administering through aerosol with other reported methods.

Author Response

  1. Cover letter with point-to-point responses is atatched.

Reviewer 2 Report

Comments and Suggestions for Authors

Comments and Suggestions for Authors are in the attached document.

Comments on the Quality of English Language

 Minor editing of English language required

Author Response

  1. Cover letter with point-to-point responses is atatched.
  2. Bi-lingual proof of the ethics committee ( AS-IACUC 21-11-1739) will be sent via email.

Round 2

Reviewer 2 Report

Comments and Suggestions for Authors

Dear authors,

I am extremely pleased that you considered some of my suggestions and criticisms and rewrote several parts of the article. This demonstrates the ability to recognize that the work has the potential to improve and that we reviewers are here to collaborate with this scientific improvement of the manuscript.

However, some questions and suggestions made by me were not made, which is the authors' right, but I need to analyze the justifications to understand why my questions and suggestions were not considered. And that's why I make them again so that you can answer me and convince me that I'm wrong or even that I didn't understand what you meant in the manuscripts.

Below I list point by point each of the items that you must answer so that I can evaluate and give my final opinion on the manuscript:

1). I did not receive the Bi-lingual proof of the ethics committee (AS-IACUC 21-11-1739). Without this document I cannot issue my opinion.

2). In line 78 of the 1st version of the manuscript I asked "Why choose this pharmacological basis?" regarding ampicillin and I was not given any response.

3). On line 98 of the 1st version of the manuscript I requested that Figure 1 be removed and wrote "It should be inserted immediately after being referenced. And in my humble opinion, figure 1 adds nothing to the scientific merit of the work. Remove it." It was not removed and I was not given an answer as to why.

4). Remove references from the 2nd revision of the manuscript in lines 111 and 113. They must appear in the body of the text as reference 13 was used, however reference 11 is not. Follow the recommendation.

5). In line 209 of the 1st version of the manuscript I requested that Figure 4 be removed and wrote "Unnecessary figure. Remove.The size and resolution of the figures are very poor. And they add nothing scientifically to the work. Just the full description is enough ." It was not removed and I was not given an answer as to why.

Author Response

Point-to-point responses and IACUC, approved by ethics committee, are atatched.

Round 3

Reviewer 2 Report

Comments and Suggestions for Authors

Again I am extremely pleased that you considered some of my suggestions and criticisms and re-wrote several parts of the article. This demonstrates the ability to recognize that the work has the potential to improve and that we reviewers are here to collaborate with this scientific improvement of the manuscript.